# Cancer risk perception and physician communication behaviors on cervical cancer and colorectal cancer screening

Diane M Harper[1,2,3]*, Madiha Tariq[4], Asraa Alhawli[4], Nadia Syed[4], Minal Patel[5,6], Ken Resnicow[6,7]

[1]Department of Family Medicine, University of Michigan School of Medicine, Ann Arbor, United States; [2]Department of Obstetrics & Gynecology, University of Michigan School of Medicine, Ann Arbor, United States; [3]Department of Women's Studies, University of Michigan, College of Literature, Science and the Arts, Ann Arbor, United States; [4]Arab Community Center for Economic and Social Services (ACCESS), Dearborn, United States; [5]Department of Health Behavior & Health Education, University of Michigan School of Public, Ann Arbor, United States; [6]Outreach and Health Disparities Research, University of Michigan Rogel Cancer Center, Ann Arbor, United States; [7]Center for Health Communications Research, University of Michigan, School of Public Health, Ann Arbor, United States

*For correspondence:
harperdi@med.umich.edu

## Abstract

**Background:** Women 50–65 years of age have the lowest cervical and colorectal cancer (CRC) screening rates among ages recommended for screening. The primary aim of this work is to determine how cancer risk perceptions and provider communication behaviors, in addition to known demographic factors, influence the uptake of both cervical and CRC screening or a single screen among women in southeast Michigan.

**Methods:** Fourteen health services and communication behavior questions were adapted from the Health Information National Trends Survey (HINTS) and administered to a multiethnic sample of adults in southeast Michigan. The outcome variable was self-reported up-to-date cervical cancer and/or CRC screening as defined by the United States Preventive Services Task Force (USPSTF). Demographic and cancer risk/communication behavior responses of the four screening populations (both tests, one test, no tests) were analyzed with multinomial regression for all comparisons.

**Results:** Of the 394 respondents, 54% were up to date for both cervical and CRC screening, 21% were up to date with only cervical cancer screening and 12% were up to date for only CRC screening. Of the 14 risk perception and communication behavior questions, only 'Did your primary care physician (PCP) involve you in the decisions about your health care as much as you wanted?' was significantly associated with women having both screens compared to only cervical cancer screening (aOR 1.67; 95% CI: 1.08, 2.57). The multivariate model showed age, and Middle East and North African (MENA) ethnicity and Black race, in addition to PCP-patient dyad decision-making to be associated with the cancer screenings women completed.

**Conclusions:** Optimizing PCP-patient decision-making in health care may increase opportunities for both cervical cancer and CRC screening either in the office or by self-sampling. Understanding the effects of age and the different interventional strategies needed for MENA women compared to Black women will inform future intervention trials aimed to increase both cancer screenings.

**Funding:** This work was supported by NIH through the Michigan Institute for Clinical and Health Research UL1TR002240 and by NCI through The University of Michigan Rogel Cancer Center P30CA046592-29-S4 grants.

## Introduction

Women 50–65 years of age have the lowest cervical cancer screening rate among those 21–65 years of age, as reported in three national surveillance databases: Behavioral Risk Factor Surveillance System (BRFSS), Health Information National Trends Survey (HINTS), and Health Center Patient Survey (HCPS) (*Harper et al., 2020*). Likewise, women 50–65 years of age also have the lowest colorectal cancer (CRC) screening rates among those 50–75 years of age (*American Cancer Society, 2020*). Only half of the women in this age group are up to date for both screens, while 25% have cervical only, 12% CRC only, and 12% have neither screen (*Harper et al., 2021a*). As we have previously reported in the evaluation of women in southeast Michigan, those more likely to be up to date for only cervical cancer screening compared to both CRC and cervical cancer screens are younger women and women of Middle East and North African (MENA) ethnicity (*Harper et al., 2021a*).

MENA women in southeastern Michigan have become an at-risk hidden population for health care as there is no US Census category that represents their ethnicity on national surveillance health surveys. The Arab immigration to the Detroit area (known as the 'Arab capital of North America' [*Crowder, 2015*] happened over the past century in four waves [*Semaan, 2015*]) in part due to homeland wars and economic poverty, with an opportunity to relocate for better employment opportunities in the auto-industry in southeastern Michigan. Early Arab immigrants were Christian and primarily from Lebanon, Israel, and Syria (*Cumoletti and Batalova, 2018*) with later increased immigration by Muslim Arabs from primarily Palestine, Iraq, and Yemen. While Arab immigration to the United States has been increasing year over year since the 1950s, health care indicators such as cervical cancer screening, incidence, or mortality have only been reported for Arab Americans after the September 11, 2001, attacks when an Arab American identity was forged (*Seeff and McKenna, 2003*; *Schwartz et al., 2004*).

Over the past two decades, cervical cancer screening rates have improved among MENA women who were significantly less likely to ever receive cervical cancer screening in their lifetime than White women (*Dallo and Kindratt, 2015*; *Yassine et al., 2010*) to the most recent report showing near equivalency with White women (*Harper et al., 2021a*; *Harper et al., 2021b*). However, while the cervical cancer screening prevalence for MENA women is similar to White women, participation in both CRC and cervical cancer screening differs for MENA women. Instead of half obtaining both cancer screenings, only 18% of MENA women are up to date for both, 50% have cervical cancer screening only, 7% CRC screening alone, and 20% report no screening at all (*Harper et al., 2021a*).

Personal risk perceptions, and cancer perceptions and knowledge as well as physician communication behaviors, are powerful influencers of screening uptake. In particular, fatalism about cancer leads to less participation in cancer screening programs among White, African American, and Hispanic race/ethnicities (*Niederdeppe and Levy, 2007*; *Chavez et al., 1997*). Greater perceptions of self-cancer risk relative to other people have been linked with an increased uptake of CRC screening (*Zhao and Nan, 2016*; *Blalock et al., 1990*, *Atkinson et al., 2015*).

In addition, physician communication behaviors may be a key modifiable determinate of cancer screening behaviors (*Peterson et al., 2016*) so much so that increasing communication behavior is included as a major objective in the *Healthy People 2030* guidelines (*HP 2030 Guidelines, 2020*). Simple physician recommendations for a prevention intervention have been successful in increasing human papillomavirus (HPV) vaccine uptake (*Kempe et al., 2019*), CRC screening (*Hudson et al., 2012*; *Atkinson et al., 2015*), and cervical cancer screening (*Luque et al., 2018*). But more specific communication patterns that examine involvement in health care decisions and dealing with women's feelings of uncertainty might better deconstruct the association with the cancer type and its related screening behaviors.

The primary aim of our study was to determine if CRC and/or cervical cancer screening uptake among the multiethnic sample of White, Black, and MENA women 50–65 years of age from southeast Michigan was associated with personal risk perceptions, cancer risk perceptions, and physician communication behavior.

## Materials and methods

### Survey measures

A cross-sectional health survey that included screening determinants questions from the NCI HINTS 5 study was developed and piloted for southeast Michigan (*Harper et al., 2021a*). Only women 50–65 years of age who answered questions about cervical cancer and CRC screening were included in this study. Questions about hysterectomy and colectomy were not included in the survey. Prior demographic descriptors significant to this screening population were included in the analyses (*Harper et al., 2021a*; *HP 2030 Guidelines, 2020*).

The respondent's perceptions of the communication behaviors of her primary care physician (PCP) were assessed with seven items adapted from the Consumer Assessment of Health Plans Study (*Marshall et al., 2001*). These include questions such as 'Did your primary care physician give you the chance to ask all the health-related questions you had?' on a four-point Likert scale (1-4): always, usually, sometimes, and never. The risk perception and knowledge section of HINTS were developed from the Health Belief Model, the Precaution Adoption Model, the Transactional Model of Stress and Coping, the Self-Regulation Model of Health Behavior, and the Protection Motivation Theory (*Vernon, 1999*; *Table 1*). The seven questions included items such as 'When I think about cancer, I automatically think about death' which were also scored on a four-point Likert scale (1-4): strongly disagree, somewhat disagree, somewhat agree, strongly agree. While these two sets of seven questions are often aggregated for a summary score, we found the detailed questions provided more insight about the screening behaviors.

For the MENA sample, the survey was distributed at 12 diverse settings of MENA interest across three southeast Michigan counties. MENA women completed the survey using online forms or paper forms, in Arabic or English, and at home or by interviewer. The remaining sample was obtained via telephone and online methods. A landline or cellphone random dial phone interview conducted by Harris Interactive Inc sampled White and African American/Black adults in the Rogel Cancer Center catchment area, which comprises about two-thirds of the state. An online sample was recruited through Dynata which oversampled African American/Black residents in the catchment area. All respondents received an incentive at completion.

### Screening outcome measures

Cervical cancer screening was considered up to date if the respondent indicated a Pap and/or HPV test within the past 3 years (*US Preventive Services Task Force et al., 2018*). For CRC screening,

**Table 1.** Survey questions.

**Cancer risk perception and knowledge**

| |
|---|
| Cancer is most often caused by a person's behavior or lifestyle |
| It seems like everything causes cancer |
| I'd rather not know my chance of getting cancer |
| There's not much you can do to lower your chances of getting cancer |
| There are so many recommendations about cancer prevention, it's hard to know which ones to follow |
| When I think about cancer, I automatically think about death |
| How likely are you to get cancer in your lifetime compared to others your age? |

**Physician communication behavior – In the past 12 months, how often did your PCP . . .**

| |
|---|
| Give you the chance to ask all health-related questions |
| Give you the attention you needed to your feelings/emotions |
| Involve you in the decisions about your health care as much as you wanted |
| Make sure you understood the things you needed to do to take care of your health |
| Explain things in a way you could understand |
| Spend enough time with you |
| Help you deal with feelings of uncertainty about your health or health care |

up to date included fecal occult blood test or fecal immunochemistry test within the past year or a colonoscopy within the past 10 years in accordance with the United States Preventive Services Task Force (USPSTF) guidelines (*US Preventive Services Task Force et al., 2016*).

## Statistical analysis

The four screening populations were defined as those women up to date with both CRC and cervical cancer screenings, with cervical cancer screening alone, with CRC screening alone or neither screening test. The demographic data were analyzed by means and frequencies. The answers to the communication behavior questions and the risk-related questions were collapsed (always/usually vs. sometimes/never) and dichotomized (strongly/somewhat disagree vs. somewhat/strongly agree) for Kruskal–Wallis analysis across the four screening populations. Multinomial logistic regression modeling was used to determine the significantly different communication and risk predictors for the four screening populations adjusting for the demographic variables. It is important to this study that comparator screenings were more than the no screening option, which has been the most commonly reported method of analysis. It is clear from the screening rates that women screen for CRC and cervical cancer differently. Allowing the analysis to include each of these screens alone as a comparator may provide insights into associations with the differential screening outcomes. Statistical analysis was completed with Statistica v 13.2 (*Dell Inc, 2016*).

## Results

There were 394 women, average age of 57.7 years (SD 4.5), who completed the CRC and cervical cancer screening questions (*Table 2*). Women declared their race/ethnicity as 69% White, 19% Black, 9% MENA, and 3% others including Asian/Pacific islander, Hispanic, American Indian, and multi-race. This distribution is very similar to the population distribution of southeastern Michigan, with some over-representation of MENA women (*Detroit-Warren-Dearborn MI Census Reporter, 2019*). The majority of the study population were Christian, had a college education, were married, had a chronic disease, and had an income between $50,000 and $99,999 also consistent with the median income for southeast Michigan (*Detroit-Warren-Dearborn MI Census Reporter, 2019*).

Religious affiliation among White women was 65% Christian (*including Chaldean, Catholic, Assyrian, Coptic, and Jehovah Witness*), 1% Muslim, 14% Jewish, and 29% no religion/no belief in god. Among Black women only two religions were reported, 77% Christian, and 23% no religion/no belief in god. Among MENA women 34% reported being Christian and 66% reported being Muslim.

Among all women, those declaring Muslim religion participated significantly less often in dual cervical cancer and CRC screening than women of other religions (p<0.05, *Supplementary file 1* – 'Screening outcomes by total population religion and by MENA ethnicity'). However, among MENA women, neither Muslim nor Christian religions were associated with preferential cervical cancer or CRC screening, with both religions having the greatest percentages in the cervical cancer-only screening group.

Moreover, women of different religions had significantly different proportions of agreement for the cancer risk perceptions and the physician communication behaviors, where Muslim women among all religions had the lowest agreement that she was involved with her physician in decisions about her health care as much as she wanted (*Supplementary file 2* – 'Agreement with cancer risk perceptions and physician communication behaviors by self-declared religion'). *Supplementary file 3* – 'Cancer screenings by religion and race/ethnicity' shows that MENA ethnicity, not a religious affiliation, is significantly associated with cervical cancer and/or CRC screening patterns.

### Cancer risk perceptions and knowledge among all women

Women in the four screening groups had significantly different proportions of agreement for cancer risk perceptions and knowledge (*Table 3*).

In particular, those with neither cancer screening had the highest agreement (strongly/somewhat) rate (33%) for the fatalism construct of 'There's not much you can do to lower your chances of getting cancer' (p=0.03 H(3,N=387) = 9.04), whereas those who had completed both CRC and cervical cancer screening had the lowest agreement rate (15%). Likewise, 'When I think about cancer, I automatically think about death' had the lowest agreement rate (48%) among those who had completed

**Table 2.** Demographic descriptors of all screening populations.

| Age, years (mean, SD) | 57.7 | 4.5 |
|---|---|---|
| Race/ethnicity | N | % |
| White | 271 | 68.8 |
| African American/Black | 75 | 19.0 |
| Other* | 11 | 2.8 |
| MENA (Middle Eastern North African) | 37 | 9.4 |
| Religion | | |
| Christian[†] | 241 | 63.6 |
| Jewish | 14 | 3.7 |
| Muslim | 26 | 6.9 |
| No religion/no belief in god | 98 | 25.9 |
| Education | | |
| High school or less | 85 | 21.6 |
| Some college | 123 | 31.2 |
| College graduate | 138 | 35.0 |
| Post college education | 48 | 12.2 |
| Marital status[‡] | | |
| Married/partnered | 236 | 59.9 |
| Single | 156 | 39.6 |
| Income | | |
| <$10,000 | 28 | 7.1 |
| $10–49,999 | 148 | 37.6 |
| $50–99,999 | 140 | 35.5 |
| ≥$100,000 | 70 | 17.8 |
| Any chronic condition[§] | 291 | 73.9 |

*Other includes Asian or other Pacific Islander, Hispanic, American Indian, and multi-race. This group was further excluded from our analyses.

[†]Christian includes Chaldean, Coptic, Assyrian, Catholic, Jehovah Witness.

[‡]Married/partnered includes married and living as married; single includes divorced, widowed, separated, and never been married.

[§]Chronic conditions include diabetes, hypertension, heart disease (heart attack, angina, congestive heart failure), lung disease (asthma, emphysema, chronic bronchitis), arthritis, depression, or cancer.

both CRC and cervical cancer screening (p=0.04 H(3,N=388)=8.22) and the highest among those who had completed only cervical cancer screening (66%).

## Communication behavior among all women

Dually screened women had significantly more positive rankings of the PCP communications of any screened population. The difference in the rankings of the PCP communication behaviors is seen in the comparator group where compared to a single screen, especially cervical cancer screening alone, women agreed significantly more often that positive PCP communication behavior was supportive for her in six of the seven questions (*Table 4*).

The second comparator group, those completing neither screen, likewise showed that the dual screening group also ranked the seventh question 'Has your PCP involved you in the decisions about your health care as much as you wanted?' significantly higher than other screening groups.

**Table 3.** Risk perceptions and cancer knowledge by screening categories.

| N=394 | Both cervical cancer and CRC screening N=213 | | Neither screening N=52 | | Cervical cancer screening alone N=82 | | CRC screening alone N=47 | |
|---|---|---|---|---|---|---|---|---|
| Cancer risk perceptions and knowledge | N | % | N | % | N | % | N | % |
| Cancer is caused by behavior or lifestyle | | | | | | | | |
| Strongly/somewhat disagree | 128 | 60.1% | 29 | 58.0% | 45 | 58.4% | 30 | 63.8% |
| Somewhat/strongly agree | 85 | 39.9% | 21 | 42.0% | 32 | 41.6% | 17 | 36.2% |
| Everything causes cancer | | | | | | | | |
| Strongly/somewhat disagree | 74 | 34.7% | 15 | 29.4% | 28 | 35.9% | 16 | 34.0% |
| Somewhat/strongly agree | 139 | 65.3% | 36 | 70.6% | 50 | 64.1% | 31 | 66.0% |
| I'd rather not know my chance of getting cancer | | | | | | | | |
| Strongly/somewhat disagree | 160 | 75.8% | 32 | 61.5% | 53 | 68.8% | 38 | 80.9% |
| Somewhat/strongly agree | 51 | 24.2% | 18 | 38.5% | 24 | 31.2% | 9 | 19.2% |
| There's not much you can do to lower your chances of getting cancer (1) | | | | | | | | |
| Strongly/somewhat disagree | 180 | 84.9% | 35 | 67.3% | 59 | 77.6% | 36 | 76.6% |
| Somewhat/strongly agree | 32 | 15.1% | 17 | 32.7% | 17 | 22.4% | 11 | 23.4% |
| There are so many recommendations about cancer prevention, it's hard to know which ones to follow | | | | | | | | |
| Strongly/somewhat disagree | 74 | 34.9% | 13 | 25.0% | 19 | 24.7% | 13 | 27.7% |
| Somewhat/strongly agree | 138 | 65.1% | 39 | 75.0% | 58 | 75.3% | 34 | 72.3% |
| When I think about cancer, I automatically think about death (2) | | | | | | | | |
| Strongly/somewhat disagree | 110 | 51.9% | 28 | 48.1% | 26 | 33.8% | 19 | 40.4% |
| Somewhat/strongly agree | 102 | 48.1% | 27 | 51.9% | 51 | 66.2% | 28 | 59.6% |
| How likely are you to get cancer in your lifetime compared to others your age? | | | | | | | | |
| Much less, less likely or about the same | 177 | 83.5% | 47 | 90.4% | 65 | 83.3% | 35 | 74.5% |
| More or much more likely | 35 | 16.5% | 5 | 9.6% | 13 | 16.7% | 12 | 25.5% |

(1) $p < 0.05$ [H(3, N=387)=9.04].
(2) $p < 0.05$ [H(3, N=388)=8.22].

## Unadjusted multinomial modeling

All cancer risk perception statements were used to model the four cancer screening behaviors. *Table 5* shows the significant associations.

Those women who had completed both CRC and cervical cancer screening compared to neither screen were significantly more likely to disagree that 'there is not much you can do to lower your chances of getting cancer' (aOR 1.65; 95% CI: 1.15, 2.35). On the other hand, the women completing both a CRC and cervical cancer screen compared to completing only the cervical cancer screen were significantly more likely to disagree that 'when they think about cancer, they automatically think about death' (aOR 1.41; 95% CI: 1.07, 2.86).

Among the PCP communication behaviors, only one behavior was significantly associated with screening outcomes in two comparator groups: 'Did your PCP involve you in the decisions about your health care as much as you wanted?' (*Table 6*).

Those women who had completed both CRC and cervical cancer screening compared to neither screen were significantly more likely to agree that 'her PCP involved her in the decisions about her health care as much as she wanted' (aOR=1.99; 95% CI: 1.26, 3.14). In addition, women who had both screens compared to only the cervical cancer screen were significantly more likely to also agree about her PCP-dyadic involvement in health care decision-making (aOR=1.72; 95% CI: 1.16, 2.55).

**Table 4.** Primary care physician communication behaviors associated with the screening categories.

| N=394 | Both cervical cancer and CRC screening N=213 | | Neither screening N=52 | | Cervical cancer screening alone N=82 | | CRC screening alone N=47 | |
|---|---|---|---|---|---|---|---|---|
| Communication behaviors | | | | | | | | |
| Chance to ask all health-related questions (1) | | | | | | | | |
| Always/usually | 195 | 92.9% | 33 | 94.3% | 62 | 82.7% | 37 | 86.1% |
| Sometimes/never | 15 | 7.1% | 2 | 5.7% | 13 | 17.3% | 6 | 14.0% |
| Give you the attention you needed to your feelings/emotions (2) | | | | | | | | |
| Always/usually | 188 | 90.8% | 31 | 86.1% | 54 | 73.0% | 30 | 73.2% |
| Sometimes/never | 19 | 9.2% | 5 | 13.9% | 20 | 27.0% | 11 | 26.8% |
| Involved you in the decisions about your health care as much as you wanted (3) | | | | | | | | |
| Always/usually | 191 | 92.7% | 29 | 76.3% | 60 | 81.1% | 35 | 83.3% |
| Sometimes/never | 15 | 7.3% | 9 | 23.7% | 14 | 18.9% | 7 | 16.7% |
| Make sure you understood the things you needed to do to take care of your health (4) | | | | | | | | |
| Always/usually | 193 | 93.7% | 36 | 94.7% | 64 | 83.1% | 37 | 86.1% |
| Sometimes/never | 13 | 6.3% | 2 | 5.3% | 13 | 16.9% | 6 | 14.0% |
| Explain things in a way you could understand | | | | | | | | |
| Always/usually | 192 | 93.7% | 33 | 86.8% | 65 | 84.4% | 38 | 88.4% |
| Sometimes/never | 13 | 6.3% | 5 | 13.2% | 12 | 15.6% | 5 | 11.6% |
| Spend enough time with you (5) | | | | | | | | |
| Always/usually | 188 | 91.3% | 31 | 86.1% | 58 | 75.3% | 34 | 79.1% |
| Sometimes/never | 18 | 8.7% | 5 | 13.9% | 19 | 24.7% | 9 | 20.9% |
| Help you deal with feelings of uncertainty about your health or health care (6) | | | | | | | | |
| Always/usually | 161 | 88.5% | 28 | 87.5% | 54 | 74.0% | 30 | 75.0% |
| Sometimes/never | 21 | 11.5% | 4 | 12.5% | 19 | 26.0% | 10 | 25.0% |

(1) p<0.05 [H (3, N=363)=7.888].
(2) p<0.001 [H (3, N=358)=17.98].
(3) p<0.01 [H (3, N=360)=12.89].
(4) p<0.05 [H (3, N=364)=9.207].
(5) p<0.01 [H (3, N=362)=13.629].
(6) p<0.05 [H (3, N=327)=10.49136].

## Multivariate multinomial modeling

When the multinomial population models were adjusted for age, race, and having any chronic disease, in addition to the cancer risk perceptions and the communication behaviors, three screening comparisons had significant predictors (*Table 7*).

None of the cancer risk perceptions were significant, nor were having any chronic diseases. The only significant predictor for women having both screens compared to neither screen was the PCP communication behavior to involve the woman as much as she wanted in her health care (PCP-patient dyad decision-making) (aOR=1.96; 95% CI: 1.22, 3.15). However, even after adjustment for the PCP-patient dyad decision-making communication style, being of MENA ethnicity, itself, was also significantly associated with less likelihood of dual vs. no screening compared to White women (aOR=0.33; 95% CI: 0.15,0.70).

Women who had both screens compared to only the cervical cancer screen were significantly older (aOR=1.13; 95% CI: 1.06, 1.21) and more involved with their PCP in decision-making (aOR=1.58; 95% CI: 1.04, 2.42). Compared to White women, Black women were significantly more

**Table 5.** Predictors of screening by cancer risk perception and knowledge.

| Cancer risk perception and knowledge | Both screens compared to neither | | | Both compared to a cervix only | | |
|---|---|---|---|---|---|---|
| | aOR | L95 | U95 | aOR | L95 | U95 |
| There's not much you can do to lower your chances of getting cancer | | | | | | |
| Strongly/somewhat disagree | **1.65** | **1.16** | **2.35** | 1.18 | 0.85 | 1.66 |
| Somewhat/strongly agree | 1.0 | | | 1.0 | | |
| When I think about cancer, I automatically think about death | | | | | | |
| Strongly/somewhat disagree | 1.0 | 0.73 | 1.36 | **1.41** | **1.07** | **1.86** |
| Somewhat/strongly agree | 1.0 | | | 1.0 | | |

Bold/red is significant.

No other screening population comparison was significant.

No other beliefs about cancer were significant.

Adjusted for all cancer risk perception and knowledge questions.

likely to have both cancer screens compared to only the cervical cancer screen (aOR=1.76; 95% CI: 1.03, 3.00) and MENA women were significantly less likely (aOR = 0.30; 95% CI: 0.15, 0.56).

Women who had only a single cervical cancer screen compared to those with a single CRC screen were significantly younger (aOR=0.84; 95% CI: 0.77, 0.93). In addition, MENA women compared to White women were significantly more likely to only screen for cervical cancer compared to CRC cancer (aOR=4.59; 95% CI: 1.13, 18.67).

## Discussion

There are significant gaps in the screening rates for both cervical cancer and CRC. We focused on understanding the many combinations of screening uptake that occur and found different predictors for dual or single CRC and cervical cancer screening among women in southeast Michigan. In prior work, having a chronic disease increased CRC or cervical cancer screening rates compared to no screen (*Harper et al., 2021a*; *Cofie et al., 2018*; *Borrayo and O'Lawrence, 2016*) and other studies have shown that cancer fatalism is a major barrier to screening uptake (*Clarke et al., 2021*; *Jun and Oh, 2013*). In this work, neither having chronic diseases nor cancer risk perceptions, such as cancer fatalism, had an influence on any cervical cancer or CRC screening behavior pattern. We did however find that positive PCP communication including the woman in her health care as much as she wants (PCP-patient dyad decision-making) was associated with completing both CRC and cervical cancer screening compared to no screen, but even more importantly also compared to only cervical cancer screening – **the first new insight from this work**.

**Table 6.** Communication behaviors predicting screening.

| Communication behavior | Both screenings compared to neither | | | Both screenings compared to cervix only | | |
|---|---|---|---|---|---|---|
| | aOR | L95 | U95 | aOR | L95 | U95 |
| Did your PCP involve you in the decisions about your health care as much as you wanted | | | | | | |
| Always/usually | **1.99** | **1.26** | **3.14** | **1.72** | **1.16** | **2.55** |
| Sometimes/never | 1.0 | | | 1.0 | | |

PCP means primary care physician.

Bold/red is significant.

No other communication behavior was significant.

No other screening population comparison was significant.

Adjusted for all communication behavior questions.

**Table 7.** Communication behavior, age, and race predict screening.

| | Both screens compared to neither | | | Both compared to cervix alone | | | Cervix only compared to CRC alone | | |
|---|---|---|---|---|---|---|---|---|---|
| | aOR | L95 | U95 | aOR | L95 | U95 | aOR | L95 | U95 |
| Age | 1.05 | 0.97 | 1.14 | **1.13** | **1.06** | **1.21** | **0.84** | 0.77 | 0.93 |
| Race/ethnicity | | | | | | | | | |
| White | 1.0 | | | 1.0 | | | 1.0 | | |
| Black | 1.74 | 0.87 | 3.46 | **1.76** | **1.03** | **3.00** | 0.49 | 0.20 | 1.24 |
| MENA | **0.33** | **0.15** | **0.70** | **0.30** | **0.16** | **0.56** | **4.59** | **1.13** | **18.67** |
| Did your PCP involve you in the decisions about your health care as much as you wanted? | | | | | | | | | |
| Always/usually | **1.96** | **1.22** | **3.15** | **1.58** | **1.04** | **2.42** | **1.01** | **0.59** | **1.73** |
| Sometimes/never | 1.0 | | | 1.0 | | | 1.0 | | |

PCP means primary care physician.
Bold/red is significant.
Adjusted for age, race/ethnicity, and PCP-patient dyad decision-making.

When one screen alone was completed, women indicated no associated involvement by the PCP in their health care decisions, whereas if there is significant PCP-patient dyad decision-making, there is a higher likelihood that both screens are completed. This creates a hypothesis that comprehensive cancer screening may best be accomplished through a PCP-patient relationship and not through targeted single cancer site screenings. To date, in the United States, specific organ cancer screenings are directed by the respective specialties. For instance, gynecologists dominate the screening method and frequency recommendations for cervical cancer screening just as gastroenterologists dominate for CRC screening. Even the USPSTF guidelines, based in primary care, consider single cancer screenings alone, without the possibility of synergistic multi-organ screenings. In addition, single cancer screenings have been the mainstay of US population programs to date with some success (*Wong and Miller, 2019*; *Pollack et al., 2020*). But our results may indicate that a single cancer screen when done outside the PCP-patient relationship is perceived more as a test to be done rather than part of general wellbeing. We hypothesize a solution of home-based or self-screening options to target dual cervical cancer (primary HPV testing) and CRC (fecal immunochemical testing with DNA) screening to be used in the PCP-patient dyad decision-making process (*El Khoury et al., 2021*; *Gorin et al., 2021*). The self-screening option could also allow the general internist who does not provide gynecologic exams the ability to be effective in this PCP-patient dyad. Moreover, we hypothesize that the PCP-patient dyad relationship may further enhance follow-up after any abnormal screenings (*Tsui et al., 2019*; *Peterson et al., 2016*).

**The second new insight** from this work is that age continues to be an important predictor of these two screens, but in opposite directions. Within the 50–65 years' age range, younger women participate more in cervical cancer screening and older women participate more in CRC screening. We hypothesize a two-level system solution: one patient-centric and one physician-centric. The patient-centric phenomenon is based in the woman's choice of her PCP. Women who choose a gynecologist as the PCP have less opportunity for dual cancer screening options, as the physician relationship is likely to be focused on the reproductive years and end at the time of menopause in her fifties, and certainly by 65 when there are no further routine reasons for gynecologic care. Likewise, we hypothesize that women who choose a general internist as the PCP are more likely to focus on other medical issues as most general internists do not provide gynecologic exams. Furthermore, we hypothesize that women who have a family physician as a PCP are the group most likely to have both screens completed as both cancer preventions are part of the foundational PCP-patient relationship of family medicine care. While it is patient-centric to choose the PCP, it is physician-centric for test ordering. We hypothesize for future work that family medicine physicians order dual screening more often than gynecologists or general internists.

**The third new insight** is the documentation of the disparities in cancer screening among MENA women which are not well reported due to US Census race/ethnicity categories. Currently race/ethnicity must be indicated by having the respondent write in their MENA heritage rather than having it

as a separate preidentified racial/ethnic category. We have shown in this work that MENA women are rarely screened for CRC, be it alone or in addition to cervical cancer screening. Past work also shows MENA women have little CRC screening (*Harper et al., 2021a*). In addition, we have shown that it is not religion itself, but rather the cultural ethnicity of MENA that is associated with the behaviors. We hypothesize that both physician and patient education in the MENA community about patient involvement in decision-making and the screens themselves will improve both cervical cancer and CRC screening.

Finally, we are the first to show for Black women in southeast Michigan that there are significant strong associations between the positive PCP-patient dyad in decision-making and the completion of both CRC and cervical cancer screening compared to cervical cancer screening alone even after adjustment for age. We hypothesize that the PCP-patient dyad for Black women occurs in patient-centered medical homes as the health care structure in which they seek care leading to dual screening.

In summary, this work shows that race/ethnicity, age, and physician communication behavior are three independent influencers associated with completing both CRC and cervical cancer screening compared to cervical cancer screening alone. This work provides a baseline for several future studies.

## Strengths and limitations
### Strengths
This study compared differential screenings for **dual cervical cancer and CRC** whereas all prior literature has only compared a single screening to no screening. In addition, these two specific internal organs (cervix and colorectum) were intentionally paired in a novel way because their barriers are quite similar: Risk factors for CRC screening in racially diverse populations include less than a high school education, low income, no health insurance, and no regular health care provider (*Larsen et al., 2017*; *Lin et al., 2021*). Other barriers to CRC screening include fatalism, religious beliefs, lack of self-worth, sexually related concerns, history of sexual abuse, and past negative experiences with screening (*Alcalá et al., 2021*; *Knight et al., 2015*). These are the same barriers that others have shown for cervical cancer screening (*Cadman et al., 2012*; *Harper et al., 2020*). This study provides the baseline data needed to explore screening uptake when both cervical cancer and CRC screenings are provided as an at-home screening option compared to in-office procedural appointments.

A **second major strength** of this study is the inclusion of MENA women who have little representation in translational clinical research. We have been able to show new deficits in CRC screening not previously identified for MENA women, establishing a baseline prevalence.

Moreover, our methodologic approach is a strength in that epidemiologic risk factors for more than one cancer screen do not exist; and those studies of a single cancer screening do not consider the association with communication and risk perception understanding. Understanding the differences for completion of one screen but not another is important to establish as prior screening strategies targeting a single screen may have missed the opportunity to present options for other cancer screenings. Finally, the population weighting previously done on this database (*Harper et al., 2021a*) provides generalizability of results for Southeast Michigan.

### Limitations
This is a cross-sectional survey implemented in the community, by Internet and phone providing a onetime sampling of self-reported health outcomes; as such, it is not possible to infer causal relationships between constructs or items in the survey. Likewise, self-reported health outcomes are not equivalent to validated medical record or claims database verification (*St Clair et al., 2017*). Health outcomes are mistakenly overestimated by about half of those who have not received the screen, whereas they are accurately reported by those who are screened (*Anderson et al., 2019*; *Bonafede et al., 2019*). An additional limitation is the self-report nature of the health descriptors. There has not been any literature indicating the veracity of self-report descriptors including health states, education, income, or other health behaviors. Both health descriptors and outcomes have been linked to social acceptability and expected responses, as well as subject to linguistic misunderstandings either from language translation or meaning of the question itself, and inaccurate recall

(*St Clair et al., 2017*). While this is a limitation, none of our covariates were significantly associated with any outcome after adjustments in the model.

Finally, our work applies to the United States and the health care structure it provides. These results may not be applicable to other national health care systems.

## Conclusions

Optimizing PCP-patient decision-making in health care may increase opportunities for both cervical cancer and CRC screening either in the office or by self-sampling. Understanding the effects of age and the different interventional strategies needed for MENA women compared to Black women will inform future intervention trials aimed to increase both cancer screenings.

## Acknowledgements

This work was supported by NIH through the Michigan Institute for Clinical and Health Research UL1TR002240 and by NCI through the University of Michigan Rogel Cancer Center P30CA046592-29-S4 grants.

## Additional information

### Competing interests

Diane M Harper: Senior editor, *eLife*. The other authors declare that no competing interests exist.

### Funding

| Funder | Grant reference number | Author |
| --- | --- | --- |
| National Cancer Institute | P30CA046592-29-S4 | Ken Resnicow |
| National Center for Advancing Translational Sciences | UL1TR002240 | Diane M Harper |

The funders had no role in study design, data collection and interpretation, or the decision to submit the work for publication.

### Author contributions

Diane M Harper, Conceptualization, Data curation, Formal analysis, Funding acquisition, Investigation, Methodology, Writing - original draft, Writing - review and editing; Madiha Tariq, Asraa Alhawli, Nadia Syed, Project administration, Writing - review and editing; Minal Patel, Conceptualization, Funding acquisition, Investigation, Methodology, Writing - review and editing; Ken Resnicow, Conceptualization, Data curation, Funding acquisition, Investigation, Methodology, Writing - review and editing

### Author ORCIDs

Diane M Harper https://orcid.org/0000-0001-7648-883X

### Ethics

Human subjects: This study was approved by Protocol #: HUM00159558 approved March 10, 2019 by the University of Michigan.

### Decision letter and Author response

Decision letter https://doi.org/10.7554/eLife.70003.sa1
Author response https://doi.org/10.7554/eLife.70003.sa2

## Additional files

### Supplementary files

• Supplementary file 1. Screening outcomes by total population religion and by Middle East and North African (MENA) ethnicity.

• Supplementary file 2. Agreement with cancer risk perceptions and physician communication behaviors by self-declared religion.

• Supplementary file 3. Cancer screenings by religion and race/ethnicity.

• Reporting standard 1. STROBE statement.

• Transparent reporting form

### Data availability

The complete raw data-set collected in the study including the participants' data set, syntax file and log files for analysis has been uploaded to https://www.openicpsr.org/openicpsr/project/148081/version/V1/view.

The following dataset was generated:

| Author(s) | Year | Dataset title | Dataset URL | Database and Identifier |
|-----------|------|---------------|-------------|-------------------------|
| Resnicow K | 2021 | Population Health Assessment in Cancer Center Catchment Areas | https://doi.org/10.3886/E148081V1 | openICPSR, 10.3886/E148081V1 |

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
