## [Decision Letter]

**Acceptance summary:**

The authors study demographic factors influencing the uptake of cervical and colorectal cancer screening among 50- to 65-years-old women. Towards those goals, they adapted and administered behavioral questions from the Health Information National Trends Survey to a multiethnic population sample in Southeast Michigan. Self-reported cancer screenings for the cancers (as defined by the USPSTF updated guidelines) served as the variable outcome. The study has public health merit in its identification of distinct predictors for cervical cancer and colorectal cancer screenings. The insights from this work on screening behavior differences among women, the perception of cancer risks and impact of positive provider communication, point to the need for exploring new ways for more holistic and integrated cancer prevention with a targeted focus.

**Decision letter after peer review:**

Thank you for submitting your article "Cancer risk perception and knowledge and physician communication behaviors: specific influences on cervical cancer and colorectal cancer screening in women 50-65 years old" for consideration by *eLife*. Your article has been reviewed by 2 peer reviewers, one of whom is a member of our Board of Reviewing Editors, and the evaluation has been overseen by Mone Zaidi as the Senior Editor. The following individual involved in review of your submission has agreed to reveal their identity: Farah Usmani (Reviewer #1).

Essential revisions:

The Referees found the work of potential interest for publication in *eLife* but have also made specific recommendations for your consideration, which are summarized as follows:

1. In a general journal and for the readers that may not be essentially familiar with the demographics in the State of Michigan, some historical context should be given for the presence of a relatively large MENA population in the region.

2. In reference to the study finding that younger women participate more in the cervical cancer screenings while the older women participate more in the colorectal cancer screenings, the authors should examine hypotheses to deconstruct this empiric age-specific and/or cancer-specific division(s). For instance, a plausible reason for decreased uptake of cervical cancer screening by older women could perhaps be the perception that they have been menopausal and symptomless, and hence would be free from reproductive tract disorders including cancers.

3. If available, the data on the religious beliefs of the participants should be included in Table 1. Also, potential correlative differences in MENA women (e.g., between Muslim versus Christian) should be addressed in the Discussion.

4. As per the stylistic specifications of *eLife*, titles of *eLife* manuscripts should not use a colon and two-part titles should also be avoided. Please revise your title with this advice in mind.

*Reviewer #1 (Recommendations for the authors):*

With reference to the finding that younger women participate more in the cervical cancer screen and older women participate more in the CRC screen, the authors may like to examine hypotheses to deconstruct the age divide in cancer screening.

A possible reason for decreased uptake of cervical cancer screening by older women could be the perception that they have been menopausal for some years now (and symptomless), and hence would be free from reproductive tract disorders including cancers.

*Reviewer #3 (Recommendations for the authors):*

In a general journal and for the readers that may not be familiar with Michigan demographics, perhaps some historical context should be given for the presence of a relatively large MENA population in the region.

The transition from a multi-ethnic population to African-American/Black and MENA is not very clear in the manuscript. Recommend some text revision in the Introduction and/or Discussion so that the non-epidemiology readers may follow it.

Would it be worth to remind GPs and family medicine providers that pelvic should be an integral part of the physical exam?

Was the role for religious beliefs investigated? For example, what were there differences, if any, between Muslin versus Christian MENA women? Otherwise a central investigational opportunity for understanding and intervention in this setting has been missing.

Could the authors speculate on the reason(s) for their finding of the age- and cancer-related in the screenings?

Ref. 1. There is a typo in the name of Dr. Sheri Seinfeld Gorin.

---

## [Author Response]

Essential revisions:The Referees found the work of potential interest for publication in eLife but have also made specific recommendations for your consideration, which are summarized as follows:1. In a general journal and for the readers that may not be essentially familiar with the demographics in the State of Michigan, some historical context should be given for the presence of a relatively large MENA population in the region.

We are happy to provide a brief historical context to this important population in Michigan. We have provided two paragraphs in the Introduction that provides a better perspective of the MENA immig<inline-graphic mime-subtype="jpeg" mimetype="image" xlink:href="media/image1.jpg" />ration in southeast Michigan

2. In reference to the study finding that younger women participate more in the cervical cancer screenings while the older women participate more in the colorectal cancer screenings, the authors should examine hypotheses to deconstruct this empiric age-specific and/or cancer-specific division(s). For instance, a plausible reason for decreased uptake of cervical cancer screening by older women could perhaps be the perception that they have been menopausal and symptomless, and hence would be free from reproductive tract disorders including cancers.

We appreciate the reviewers' interest in this age dichotomy within our internal age range. This is a first notification of this health gap that we are happy to have established in the literature. Our Discussion section has been rewritten to include the third paragraph which is devoted to the hypothesis generating questions that come from our results.

In the US, our first hypothesis is that this phenomenon is driven from the provider side, not the patient side. Hence, gynecologists, general internists and family medicine physicians are the front line physicians for cancer screenings.

Women in the US have the opportunity to declare any one of these specialties as their Primary Care Physician (PCP).

Our hypothesis is that the woman stays with her gyn if she started early in her reproductive life with a gyn as her PCP, and the gyn is focused on cervical cancer screening through the age of 65 where it is no longer recommended. Likewise, the woman who chooses a general internist for her PCP usually is referred to a gyn for the pelvic exam as this is not in the scope of practice for most general internists — nor is their office equipped for the requirements for chaperoning, and the equipment needs of a table with stirrups, specula and sampling devices. In addition, as women gather new medical diseases, such as hypertension, diabetes, COPD, rheumatoid arthritis, she is more likely to switch to the general internist providing therapeutic care as her current PCP. This is often around the age of 50, where the focus of the health care is on the already established diseases and not on screening for new cancers, especially not those that require a speculum exam.

On the other hand, family medicine physicians practice the full scope of care and are in the position to provide both gyn and GI cancer screenings. Our next project will be to evaluate the provider type who orders the screening tests from the large claims databases such as Medicare or Optum to see where the correlation between physician type and dual or single cancer screens occurs.

3. If available, the data on the religious beliefs of the participants should be included in Table 1. Also, potential correlative differences in MENA women (e.g., between Muslim versus Christian) should be addressed in the Discussion.

We are grateful for this suggestion as did collect this information and were able to reconstruct the Results section with this perspective.

Three supplementary tables have been added to the Results section. Supplementary table I shows that while overall Muslims screen differently than all other religions, the MENA women do not show differences in screening by Muslim vs Christian religion. We describe in the 4^th^ paragraph of the Discussion section that this reflects on the culture of MENA being more important than the religion.

Supplemental table 2 shows the multivariate multinomial modeling of religion vs.

race/ethnicity to predict dual, single or neither screening. This table shows that screening is not associated with religion when the model is adjusted for both religion and race/ethnicity. This supports the 4^th^ paragraph of the Discussion section.

Supplemental table 3 shows the differences in cancer risk perception and physician communication behaviors by religion. This table shows that Christians and Jewish women have a significantly higher agreement that there is not much you can do to lower your chances of getting cancer than do Muslim women. Similarly, when the Jewish women think about cancer, a large portion automatically think about death, es eciall com ared to Muslim women. Finally, Muslim women are the least to be involved with their PCP in a dyad decision making relationship. These results are incorporated into Table 6 where adjustments for religion, race, age and all univariate predictors were made. The results of Table 6 show that the culture of MENA women overrides the religion in significant prediction for cancer screenings.

4. As per the stylistic specifications of eLife, titles of eLife manuscripts should not use a colon and two-part titles should also be avoided. Please revise your title with this advice in mind.

We have changed the title of the manuscript to be in accordance with *eLife* stylistic policy.

Reviewer #3 (Recommendations for the authors):In a general journal and for the readers that may not be familiar with Michigan demographics, perhaps some historical context should be given for the presence of a relatively large MENA population in the region.

See response to Essential revisions #1.

The transition from a multiethnic population to African-American/Black and MENA is not very clear in the manuscript.

We have rewritten the manuscript to clarify that our study population is comprised of White, Black and MENA women only.

Recommend some text revision in the Introduction and/or Discussion so that the non-epidemiology readers may follow it.Would it be worth to remind GPs and family medicine providers that pelvic should be an integral part of the physical exam?

We appreciate this thought, but evidence has evolved to show that the sensitivity and specificity of a pelvic exam does not provide any information to the PCP that would be diagnostic, hence the practice of pelvic exams routinely has been abandoned.

Was the role for religious beliefs investigated? For example, what were there differences, if any, between Muslin versus Christian MENA women? Otherwise a central investigational opportunity for understanding and intervention in this setting has been missing.

Including the variable of religion into the analysis has been very important to be able to state clearly that the results are culturally and ethnically derived, not by religion.

Could the authors speculate on the reason(s) for their finding of the age- and cancer-related in the screenings?

We agree that these two aspects of our results provide us with several hypothesis-generating studies for the future. We have hypothesized that the age dichotomy is generated from the woman's characteristics — if she has chosen a gyn as a PCP she is likely to get a cervical cancer screen in her 50's but the patient will be gone to her practice by the time she is 65; similarly if the woman has chosen a general internist for her PCP, it is unlikely that she will have a cervical cancer screen between 50 and 65 but will have a CRC screen.

Likewise we hypothesized that the dual screening vs single vs no screenings comes from the type of physician ordering the test rather than by a patient's characteristics, as a general internist is unlikely to remember to refer out for the cervical cancer screening and cannot do it in the office. We hypothesize that home based cervical and CRC screenings would alleviate the issue of in office screening.

Ref. 1. There is a typo in the name of Dr. Sheri Seinfeld Gorin.

Dr. Gorin confirmed the spelling of her name in the reference section.